# Immune suppression is associated with enhanced systemic inflammatory, endothelial and procoagulant responses in critically ill patients

**Xanthe Brands** [1‡](ORCID)*, **Fabrice Uhel**[1‡], **Lonneke A. van Vught**[1], **Maryse A. Wiewel**[1], **Arie J. Hoogendijk**[1], **René Lutter**[2], **Marcus J. Schultz**[3,4,5], **Brendon P. Scicluna**[1,6,7,8], **Tom van der Poll**[1,9]

1 Center for Experimental and Molecular Medicine (CEMM), Amsterdam University Medical Centers—Location AMC, University of Amsterdam, Amsterdam, The Netherlands, 2 Respiratory Medicine and Experimental Immunology, Amsterdam University Medical Centers—Location AMC, Amsterdam, The Netherlands, 3 Department of Intensive Care & Laboratory of Experimental Intensive Care and Anesthesiology, Amsterdam University Medical Centers, Location Academic Medical Center, University of Amsterdam, Amsterdam, The Netherlands, 4 Mahidol Oxford Tropical Medicine Research Unit, Bangkok, Thailand, 5 Nuffield Department of Medicine, University of Oxford, Oxford, United Kingdom, 6 Department of Clinical Epidemiology, Biostatistics and Bioinformatics, Amsterdam University Medical Centers—Location AMC, University of Amsterdam, Amsterdam, The Netherlands, 7 Department of Applied Biomedical Science, Faculty of Health Sciences, Mater Dei Hospital, University of Malta, Msida, Malta, 8 Centre for Molecular Medicine and Biobanking, University of Malta, Msida, Malta, 9 Division of Infectious Diseases, Amsterdam University Medical Centers—Location AMC, University of Amsterdam, Amsterdam, The Netherlands

‡ XB and FU share first authorship on this work.
* x.brands@amsterdamumc.nl

**Data Availability Statement:** All relevant data are within the paper and its Supporting Information files.

## Abstract

### Objective

Patients admitted to the Intensive Care Unit (ICU) oftentimes show immunological signs of immune suppression. Consequently, immune stimulatory agents have been proposed as an adjunctive therapy approach in the ICU. The objective of this study was to determine the relationship between the degree of immune suppression and systemic inflammation in patients shortly after admission to the ICU.

Design: An observational study in two ICUs in the Netherlands.

### Methods

The capacity of blood leukocytes to produce cytokines upon stimulation with lipopolysaccharide (LPS) was measured in 77 patients on the first morning after ICU admission. Patients were divided in four groups based on quartiles of LPS stimulated tumor necrosis factor (TNF)-α release, reflecting increasing extents of immune suppression. 15 host response biomarkers indicative of aberrations in inflammatory pathways implicated in sepsis pathogenesis were measured in plasma.

### Results

A diminished capacity of blood leukocytes to produce TNF-α upon stimulation with LPS was accompanied by a correspondingly reduced ability to release of IL-1β and IL-6. Concurrently

**Funding:** This research was performed within the framework of the Center for Translational Molecular Medicine (CTMM) (www.ctmm.nl), project Molecular Diagnosis and Risk Stratification of Sepsis (grant 04I-201). The sponsor CTMM was not involved in the design and conduction of the study; nor was the sponsor involved in collection, management, analysis, and interpretation of the data or preparation, review or approval of the article. Decision to submit the article was not dependent on the sponsor. X.B. was supported by a grant from the Netherlands Organization for Health Research and Development (ZonMW #50-53000-98-139).

**Competing interests:** The authors have declared that no competing interests exist.

measured plasma concentrations of host response biomarkers demonstrated that the degree of reduction in TNF-α release by blood leukocytes was associated with increasing systemic inflammation, stronger endothelial cell activation, loss of endothelial barrier integrity and enhanced procoagulant responses.

## Conclusions

In patients admitted to the ICU the strongest immune suppression occurs in those who simultaneously display signs of stronger systemic inflammation. These findings may have relevance for the selection of patients eligible for administration of immune enhancing agents.

## Trial registration

ClinicalTrials.gov identifier NCT01905033.

## Introduction

Critical illness is associated with a disturbed homeostasis characterized by a complex interplay between hyperinflammation and immune suppression [1–3]. Exaggerated proinflammatory responses include an excessive systemic release of inflammatory cytokines, endothelial cell activation and dysfunction, and activation of the coagulation system. Conversely, the reduced capacity of blood leukocytes to produce pro-inflammatory cytokines upon stimulation with lipopolysaccharide (LPS) has been described as a common feature of immune suppression [2, 4]. These host response aberrations have been reported in various intensive care conditions including sepsis, surgery and trauma patients [1–4]. Originally, hyperinflammation and immune suppression were considered subsequent phases in the immune response to critical illness, and the term "compensatory anti-inflammatory response syndrome" was introduced for the (later) immune suppressive "phase" [5, 6]. However, more recent evidence supports the co-existence of these seemingly opposite responses in patients at admission to the intensive care unit (ICU), although the extent of this association still needs to be determined [1–3].

In the past decades multiple clinical trials evaluating immune modulatory agents have been conducted in critically ill patients, particularly in those with sepsis [2, 7, 8]. Partially driven by the failure of these trials to show benefit, controversy has grown over how the host response should be manipulated in critically ill patients. In this context immune profiling may guide therapeutic options in the future, with selection of patients with predominantly exaggerated systemic inflammation for anti-inflammatory therapies and selection of those with dominant immune suppressive features for immune stimulating strategies [2, 9]. As an example, diminished HLA-DR expression on circulating monocytes and a reduced capacity of blood leukocyte to produce TNF-α upon LPS stimulation have been used as markers of immune suppression for patient selection and treatment monitoring in studies evaluating the immune enhancing effects of recombinant interferon-γ and granulocyte-macrophage colony stimulation factor in sepsis [10–12]. To date, evidence for the effectiveness of such precision strategies is scarce.

We here hypothesized that the extent of immune suppression is associated with the degree of hyperinflammation in patients with critical illness. We considered testing this hypothesis relevant considering that evidence supporting this would hamper selection of patients for targeted anti-inflammatory or immune enhancing therapies. To this end, we used the decreased

capacity of whole blood leukocytes to produce TNF-α in response to LPS-stimulation as a readout for critically-ill patient immune suppression in conjunction with measurement of a comprehensive set of plasma biomarkers reflecting a variety of systemic pro-inflammatory responses linked to specific pathophysiological mechanisms, focusing on cytokine release, endothelial cell activation and activation of the coagulation system. Part of this work has been presented during the French Intensive Care Society International Congress 2021 [13].

## Methods

### Study population and sample collection

Consecutive patients older than 18 years admitted to the ICU in the Academic Medical Center (Amsterdam, the Netherlands) between April 2012 and June 2013 were included when they had at least two systemic inflammatory response syndrome criteria upon admission (body temperature $\leq 36°C$ or $\geq 38°C$, tachycardia >90/min, tachypnea >20/min or $pCO_2$ <4.3 kPa, leukocyte count $< 4.10^9$/L or $>12/10^9$/L) [14]. Patients transferred from another ICU, receiving antibiotics for more than 48 hours before admission, and/or with an expected length of ICU stay of less than 24 hours were excluded. The presence of an infection was assessed by attending physicians, and the likelihood of infection was subsequently adjudicated by independent observers using a four point scale (ascending from *none*, *possible*, *probable* to *definite*) [15]. Sepsis was defined as the presence of an infection diagnosed within 24 hours after admission with a *possible*, *probable* or *definite* likelihood combined with at least one general, inflammatory, hemodynamic, organ dysfunction or tissue perfusion variables derived from the 2001 International Sepsis Definitions Conference [16]. Patients without infection upon admission, or patients initially suspected of infection but with a *post hoc* infection likelihood of *none* were classified as non-septic critically ill patients. Healthy subjects serving as controls for whole blood stimulation results were matched with regard to age, sex, and timing of blood draw. The study received approval from the medical ethical committee of the Academic Medical Center in Amsterdam (no. NL 34294.018.10), and was registered at the Central Committee for Human Research. Written informed consent to participate in the study and for publication was obtained from all patients (or legal representative) and healthy controls.

### Clinical variables

Sequential Organ Failure Assessment (SOFA) [17] and Acute Physiology And Chronic Health Evaluation (APACHE) IV scores [18] were calculated upon ICU admission. Shock was defined by the need of vasopressors for hypotension at a dose of at least 0.1 µg/kg/min during at least 50% of the ICU day. Comorbidities were defined as described [19] and the Charlson comorbidity index [20] was calculated based hereon. Acute respiratory distress syndrome (ARDS) and acute kidney injury (AKI) were defined according to strict definitions [21, 22].

### Whole blood stimulation and biomarker assays

Blood was obtained at 9:00 AM on the first day after admission to the ICU. Within two hours after collection, heparin-anticoagulated whole blood was stimulated for 3 hours at 37°C with 5% $CO_2$ and 95% humidity in pyrogen-free RPMI 1640 medium (Life Technologies, Bleiswijk, the Netherlands) with or without 100 ng/mL ultrapure LPS (*Escherichia coli* 0111:B4; 100 ng/mL, InvivoGen, Toulouse, France). After stimulation, supernatants were collected and stored at -80°C until measurement of tumor necrosis factor (TNF)-α, interleukin (IL)-6 and IL-1β (assays described below). Blood stimulation experiments were partly reported in an earlier publication from our group [22]. Additionally, EDTA anticoagulated blood was obtained for

measurements in plasma. The following assays were used: TNF-α, IL-1β, IL-6, IL-8, IL-10, soluble intercellular adhesion molecule-1 (sICAM-1), and soluble (s)E-selectin were measured by FlexSet cytometric bead array (BD Biosciences, San Jose, CA) using a FACS Calibur (Becton Dickinson, Franklin Lakes, NJ); angiopoietin-1, angiopoietin-2, matrix metalloproteinase (MMP)-8, antithrombin (R&D Systems, Abingdon, UK), protein C and D-dimer (Procartaplex, eBioscience, San Diego, CA) were measured by Luminex multiplex assay using a BioPlex 200 (BioRad, Hercules, CA). Platelet counts were determined by hemocytometry, prothrombin time (PT) and activated partial thromboplastin time (aPTT) by using a photometric method with Dade Innovin Reagent or by Dade Actin FS Activated PTT Reagent, respectively (Siemens Healthcare Diagnostics). C-reactive protein (CRP) was determined by immunoturbidimetric assay (Roche diagnostics). Leukocyte counts and differentials were determined by fluorescence flow cytometry on a Sysmex® XN9000 analyser (Sysmex Corporation, Kobe, Japan). Normal biomarker values were obtained from age- and gender-matched healthy volunteers, with the exception of CRP, PT and aPTT (routine laboratory reference values).

## Statistical analyses

A formal sample size calculation was not done prior to the study (to the best of our knowledge previous studies associating whole blood leukocyte stimulations with biomarkers of systemic inflammation have not been performed). Patients were stratified into quartiles based on the capacity of their blood leukocytes to produce TNF-α. Data distribution was assessed using histograms and Shapiro-Wilk tests. Non-normally distributed continuous variables are presented as median and interquartile range (IQR, 25th, 75th percentile) and were analyzed with Kruskal-Wallis test followed by Dunn's post-test of multiple comparisons using rank sums. Categorical variables, presented as numbers (percentages), were analyzed using Chi-square test or Fisher's exact test when appropriate. Correlations were measured using Spearman's rank correlation test. Analyses were performed in R (v 3.5.1, R Foundation for Statistical Computing, Vienna, Austria). Multiple-comparison-adjusted $P$ values less than 0.05 defined significance.

## Results

### Stratification of ICU patients according to TNF-α production capacity and clinical outcome

Between April 2012 and June 2013, 77 critically ill patients and 19 age- (median, 63 years [IQR, 52–71 years]) and sex-matched (39% male) healthy volunteers were included. 51 (66%) patients had sepsis upon admission (for admission diagnoses see S1 Table).

In order to evaluate the extent of immune suppression in critically ill patients, we measured the cytokine production capacity of whole blood leukocytes upon *ex vivo* stimulation with LPS. Blood leukocytes of ICU patients produced less pro-inflammatory TNF-α, IL-1β and IL-6 after LPS stimulation compared with blood leukocytes from healthy volunteers (S1 Fig). We hypothesized that critically ill patients with increasing severity of immune suppression concurrently show stronger systemic proinflammatory responses. Given that a reduced TNF-α production capacity by blood leukocytes has been widely recognized as a hallmark feature of immune suppression in critically ill patients [6, 23, 24], we stratified patients into four groups based on quartiles of LPS-induced TNF-α production. The quartile with the highest TNF production capacity (>896 pg/ml; n = 19) did not differ from healthy subjects and is further referred to as "normal"; the other quartiles are further indicated as "slightly reduced" (TNF-α 384–896 pg/ml; n = 19), "moderately reduced" (TNF-α 128–383 pg/ml, n = 19) and "strongly reduced" (TNF-α <128 pg/ml; n = 20; Fig 1A). Reduction in TNF-α production capacity was

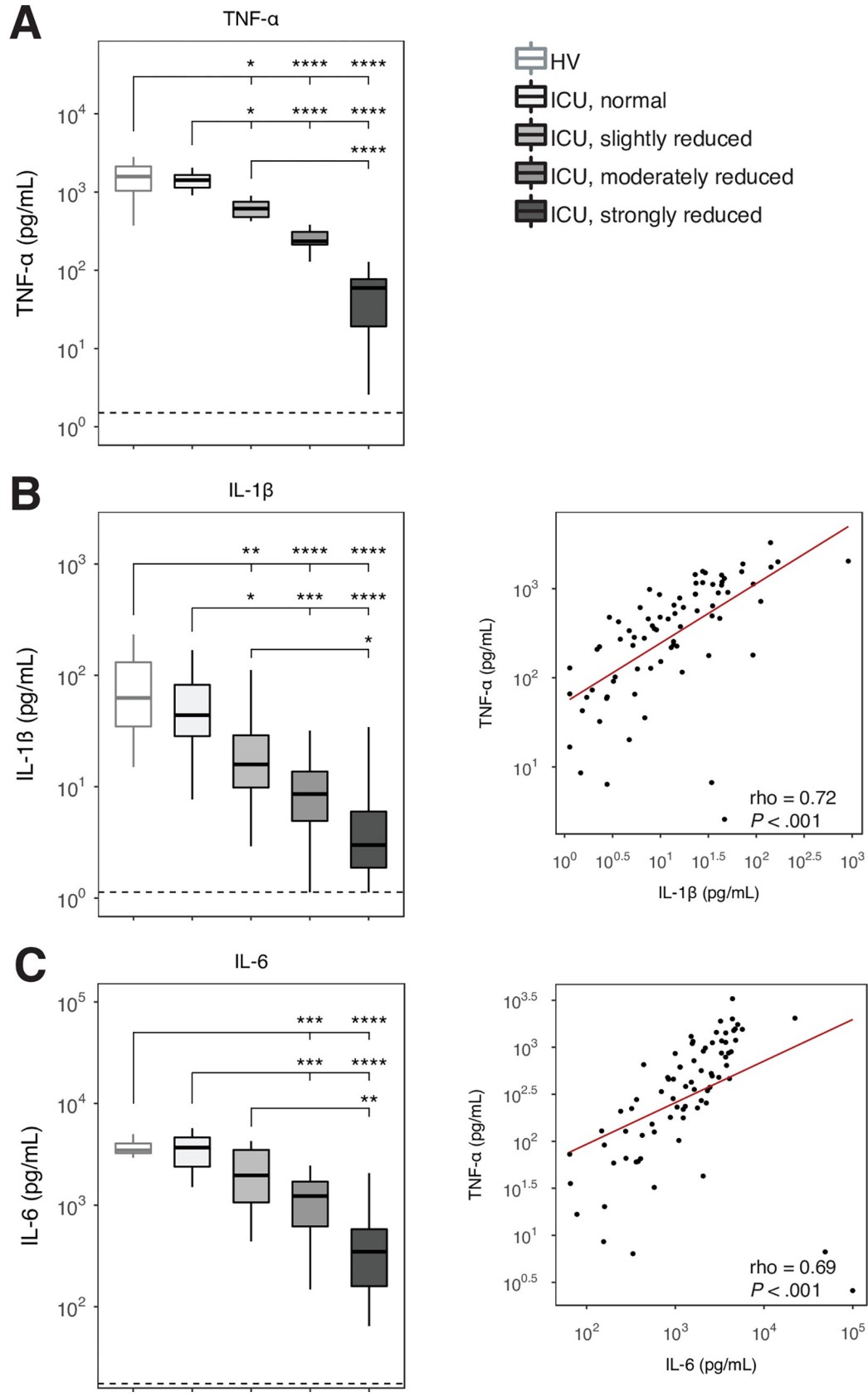

**Fig 1. Whole-blood leukocyte responsiveness to LPS in critically ill patients. (A)** LPS-induced whole blood leukocyte cytokine production in critically ill patients on the first day after admission (n = 77) stratified according to quartiles of TNF-α production capacity (normal, slightly reduced, moderately reduced, and strongly reduced), and in age and sex-matched healthy controls (n = 19). Dotted lines indicate median cytokine concentrations in unstimulated control samples. Data are expressed as box and whisker diagrams as specified by Tukey. HV, healthy volunteers; ICU, critically ill patients. $^*P < 0.05$, $^{**}P < 0.01$, $^{***}P < 0.001$, $^{****}P < 0.0001$. Dot plots depicting the relationship between LPS-induced TNF-α and **(B)** IL-1β, and **(C)** IL-6 whole blood production capacity in critically ill patients. Rho, Spearman's correlation coefficient.

associated with a proportionally reduced release of IL-1β and IL-6 upon stimulation with LPS (**Fig 1B and 1C**), and TNF-α levels measured in LPS stimulated blood of patients strongly correlated with IL-1β (rho = .72 P< .001) and IL-6 levels (rho = .69, P< .001) detected in supernatants, suggesting that the stratification of patients based on TNF-α production capacity of blood leukocytes resulted in conditions of increasing degrees of immune suppression.

Patients stratified according to TNF-α production capacity did not differ in terms of demographics, chronic comorbidities or severity of disease (**Table 1**). Among patients admitted for

**Table 1. Baseline characteristics and outcomes of patients stratified according to whole blood TNF-α production capacity upon LPS stimulation.**

| | Normal (n = 19) | Slightly reduced (n = 19) | Moderately reduced (n = 19) | Strongly reduced (n = 20) | P value |
|---|---|---|---|---|---|
| TNF-α (range), pg/mL | > 896 | 384–896 | 128–383 | <128 | |
| Demographics | | | | | |
| Age, years | 65 [57–73] | 67 [52–78] | 57 [46–65] | 57 [48–64] | 0.06 |
| Male sex | 14 (73.7) | 12 (63.2) | 13 (68.4) | 7 (35.0) | 0.07 |
| BMI, kg/m$^2$ | 25.3 [23.6–31.6] | 26.1 [24.0–28.6] | 24.9 [23.1–28.5] | 24.9 [23.1–27.0] | 0.79 |
| Race, white | 15 (78.9) | 17 (89.5) | 17 (89.5) | 15 (75.0) | 0.57 |
| Medical admission | 6 (31.6) | 3 (15.8) | 5 (26.3) | 5 (25.0) | 0.76 |
| Sepsis admission diagnosis | 11 (57.9) | 8 (42.1) | 14 (73.7) | 18 (90.0)† | 0.011 |
| Chronic comorbidities | | | | | |
| None | 3 (15.8) | 7 (36.8) | 5 (26.3) | 5 (25.0) | 0.56 |
| Charlson comorbidity index | 3 [2 – 4] | 4 [2 – 5] | 3 [1 – 5] | 3 [1 – 5] | 0.88 |
| Severity at time of admission to ICU | | | | | |
| APACHE IV score | 79 [67–94] | 76 [58–100] | 76 [64–91] | 75 [61–103] | 0.99 |
| Acute physiology score | 68 [49–84] | 52 [38–89] | 70 [58–78] | 61 [51–97] | 0.74 |
| SOFA score | 5 [4 – 8] | 8 [6 – 9] | 8 [6 – 9] | 8 [6 – 10] | 0.23 |
| Shock | 7 (36.8) | 7 (36.8) | 10 (52.6) | 14 (70.0) | 0.13 |
| ARDS | 3 (15.8) | 3 (15.8) | 2 (10.5) | 9 (45.0) | 0.06 |
| AKI | 5 (26.3) | 9 (47.4) | 9 (47.4) | 7 (35.0) | 0.47 |
| Leukocyte counts and differentials | | | | | |
| WBC max, x10$^9$/L | 14.90 [10.85–17.50] | 12.40 [9.65–15.80] | 13.40 [10.20–19.30] | 14.20 [9.98–18.80] | 0.82 |
| Neutrophils, x10$^9$/L | 10.12 [7.12–14.05] | 9.61 [7.39–11.64] | 8.60 [7.12–11.12] | 8.79 [7.14–14.49] | 0.87 |
| Monocytes, x10$^9$/L | 0.89 [0.53–1.10] | 0.66 [0.46–0.86] | 0.56 [0.44–0.85] | 0.38 [0.24–0.56] | 0.05 |
| Lymphocytes, x10$^9$/L | 1.05 [0.57–1.45] | 0.89 [0.72–1.16] | 0.84 [0.44–1.29] | 0.84 [0.74–1.67] | 0.67 |
| Outcome | | | | | |
| ICU length of stay, days | 5 [4 – 8] | 3 [3 – 11] | 4 [3 – 7] | 6 [4 – 9] | 0.48 |
| ICU mortality | 3 (15.8) | 4 (21.1) | 2 (10.5) | 5 (25.0) | 0.76 |

Data presented as median [interquartile range], or n (%). Continuous variables were compared using the Kruskall-Wallis test. Associations between categorical variables were tested using the Fisher's exact test. *P* values represent comparisons between the four groups.

Abbreviations: AKI, acute kidney injury; APACHE, Acute Physiology and Chronic Health Evaluation; ARDS, acute respiratory distress syndrome; SOFA, Sequential Organ Failure Assessment; WBC, white blood cell count.

a sepsis (n = 51) or for a non-infectious diagnosis (n = 26), 40 (78.4%) and 18 (69.2%) showed reduced TNF production capacity (≤896 pg/mL), respectively. A sepsis admission diagnosis was over-represented in patients with a strongly reduced TNF-α production-capacity. White blood cell and neutrophil counts did not differ between groups; patients with the lowest TNF-α production capacity had the lowest monocyte numbers in peripheral blood. TNF-α production adjusted for monocyte counts remained significantly lower in these patients (**S2 Fig**). The ICU length of stay and ICU mortality did not differ between patient groups.

## A reduced TNF-α production capacity is associated with enhanced systemic inflammatory responses

In order to obtain insight into the association between blood leukocyte responsiveness and systemic proinflammatory host responses, we compared the levels of 15 plasma biomarkers reflecting major pathways involved in the pathogenesis of critically illness between patients stratified according to quartiles of TNF-α production capacity. When compared with control subjects, all critically ill patients showed signs of a dysregulated host response on ICU admission, with elevated levels of proinflammatory (CRP, IL-6, IL-8, MMP-8) and anti-inflammatory (IL-10) mediators (**Fig 2**). Patients with moderately to strongly reduced TNF-α production-capacity showed enhanced systemic pro- and anti-inflammatory host responses

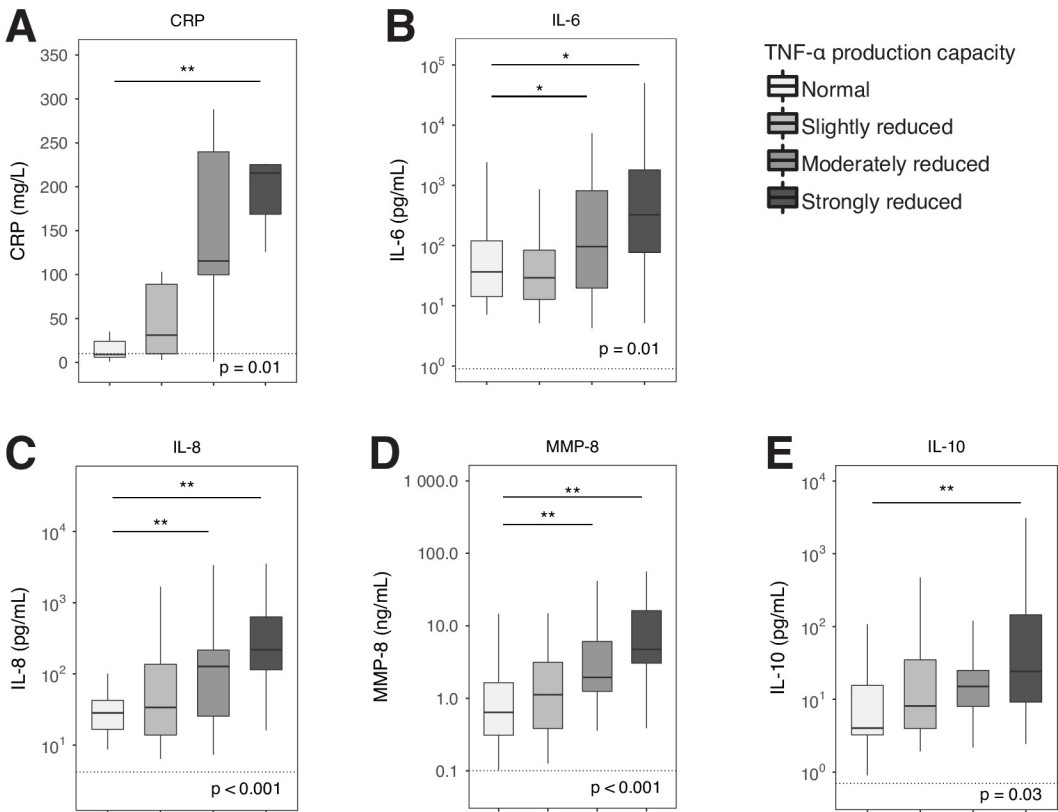

**Fig 2. Biomarkers of systemic inflammatory responses in critically ill patients stratified according to whole blood TNF-α production capacity.** Data are presented as box and whiskers, as specified by Tukey. Dotted lines represent median values obtained in age-matched healthy subjects. Comparisons between groups were performed using the Kruskall-Wallis test followed by Dunn's post-tests adjusted for multiple comparisons (Bonferroni). * P < .05, ** P < .01. CRP, C-reactive protein; IL, interleukin; MMP, matrix metalloproteinase; TNF, tumor necrosis factor.

compared with those with a normal TNF-α production capacity. These data suggest that critically ill patients with the strongest immunosuppression (lowest TNF-α production capacity) concurrently show stronger systemic inflammatory responses.

### A reduced TNF-α production capacity is associated with enhanced endothelial cell activation and loss of vascular integrity

We measured biomarkers for endothelial cell activation (plasma levels of sICAM-1 and sE-S-electin) and vascular integrity (angiopoietin 1 and 2) on admission to the ICU (**Fig 3**). Patients with a strongly reduced leukocyte TNF-α production capacity displayed the highest levels of sICAM-1, angiopoeitin-2 and angiopoietin-2:1 ratio, indicative of stronger endothelial cell activation and a more profound loss of vascular integrity.

### A reduced TNF-α production capacity has some association with enhanced procoagulant responses

We measured biomarkers of coagulation activation (D-dimer, PT, aPTT, platelet counts) and anticoagulant mechanisms (protein C, antithrombin) on admission to the ICU (**Fig 4**). Patients with a moderately and strongly reduced TNF-α production-capacity showed increased plasma levels of D-dimer and decreased levels of antithrombin respectively, indicative of a more disturbed hemostatic balance.

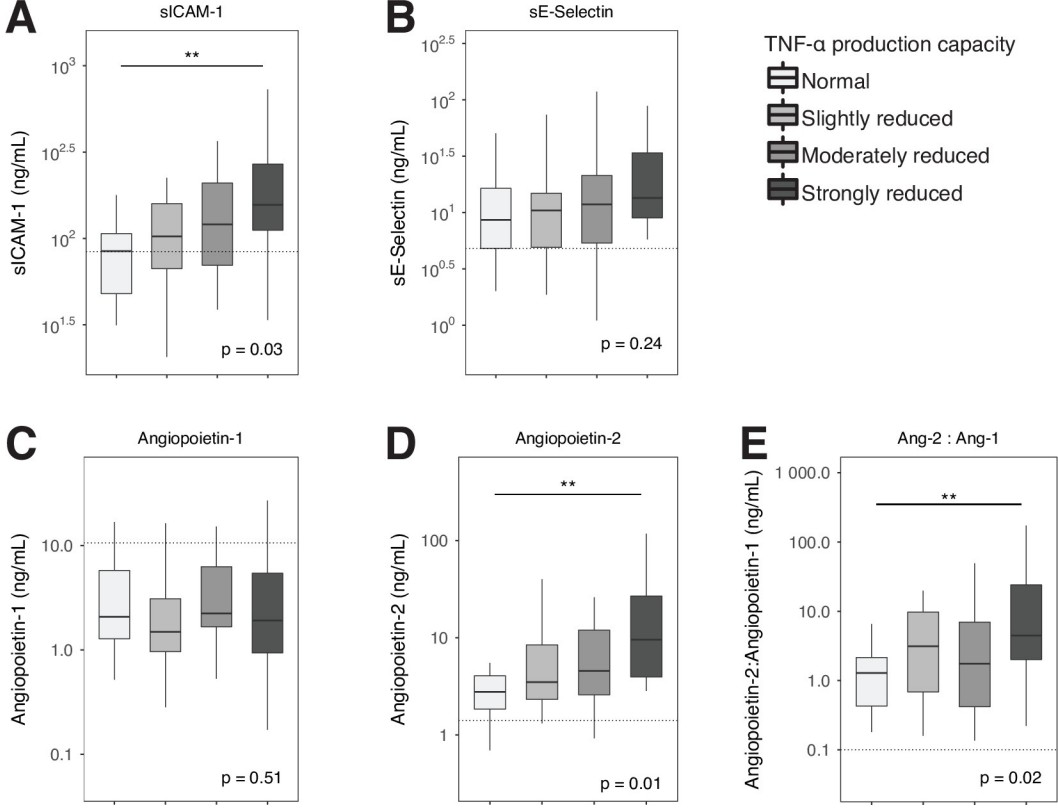

**Fig 3. Endothelial cell activation biomarkers in critically ill patients stratified according to whole blood TNF-α production capacity.** Data are presented as box and whiskers, as specified by Tukey. Dotted lines represent median values obtained in age-matched healthy subjects. Comparisons between groups were performed using the Kruskall-Wallis test followed by Dunn's post-tests adjusted for multiple comparisons (Bonferroni). ** P < .01. ANG, angiopoietin; sE-Selectin, soluble E-selectin; sICAM, soluble intercellular adhesion molecule; TNF, tumor necrosis factor.

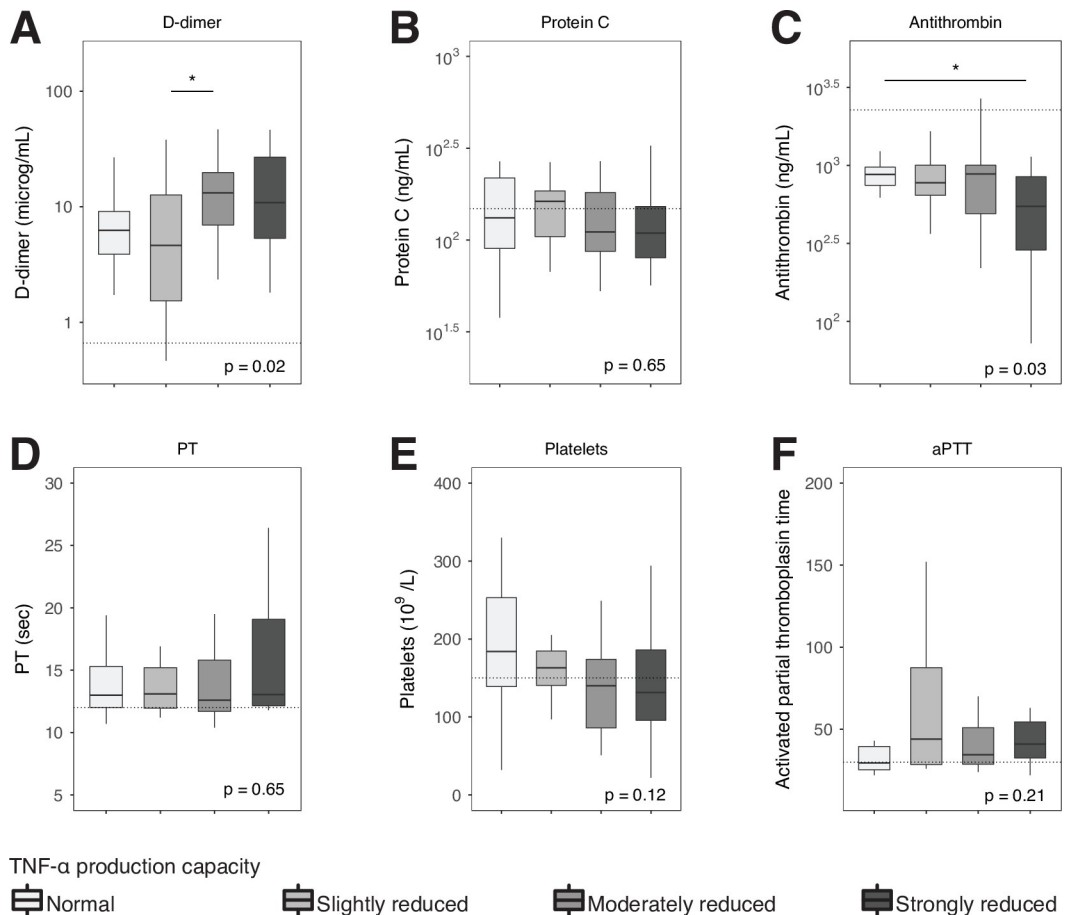

**Fig 4. Coagulation activation biomarkers in critically ill patients stratified according to whole blood TNF-α production capacity.** Dotted lines represent median values obtained in age-matched healthy subjects. Comparisons between groups were performed using the Kruskall-Wallis test followed by Dunn's post-tests adjusted for multiple comparisons (Bonferroni). * P < .05. aPTT, activated partial thromboplastin time; PT, prothrombin time; TNF, tumor necrosis factor.

## Discussion

Immune suppression is a common feature in critically ill patients and administration of immune stimulatory agents has been advocated as a new therapeutic strategy to reverse this host response aberration in this population. However, drugs that stimulate the immune system may enhance excessive proinflammatory responses also present in patients selected for this adjunctive therapy. We here sought to obtain insight in the proportionality of immune suppression and concurrently detectable systemic hyperinflammation in critically ill patients. To this end we used the TNF-α production capacity of LPS-stimulated blood leukocytes as a read-out for immune suppression, stratified patients into quartiles according to the extent in which this response was impaired and measured 15 biomarkers indicative of dysregulation of proinflammatory host response mechanisms in plasma. We demonstrate that critically ill patients with the most severe immunosuppression (as indicated by the lowest TNF-α production capacity) concurrently show the strongest signs of systemic inflammatory and endothelial responses.

Immune suppression is considered an important determinant in the outcome of critical illness [1–3, 24, 25]. Previous studies also used a reduced capacity of blood leukocytes to produce

proinflammatory cytokines upon *ex vivo* stimulation with bacterial agonists like LPS in patients with sepsis or non-infectious critical illness [2, 4, 6, 10, 11, 24, 26]. Measurement of HLA-DR expression on monocytes is another commonly used readout for immune suppression in clinical settings [2, 6, 10–12, 24, 27]; monocyte HLA-DR levels showed a strong correlation with the responsiveness of whole blood leukocytes to LPS in critically ill patients [10, 26, 28]. Likewise, in a model of in vitro LPS tolerance a reduced ability of monocytes to produce TNF-α was associated with a diminished HLA-DR expression [29]. These data provide further validity to the use of TNF-α production capacity of blood leukocytes to stratify patients in groups with different severities of immune suppression. Moreover, low TNF-α producers also exhibited reductions in IL-1β and IL-6 release in LPS-stimulated whole blood, suggesting that these patients indeed displayed a stronger immunosuppressive phenotype.

We measured TNF-α, IL-1β and IL-6 levels after a 3-hour incubation of whole blood with LPS. Likely, monocytes are the main producers of cytokines in this setting. Patients with the lowest TNF-α production capacity showed a clear trend toward lower monocyte numbers in blood. However, strong differences between quartiles based on whole blood TNF-α production capacity remained after adjustment for monocyte counts, suggesting that an altered responsiveness of monocytes and not their numbers was responsible for the immunosuppressive phenotype. This notion is supported by previous studies showing a reduced capacity of blood monocytes harvested from critically ill sepsis patients to activate nuclear factor-κB and to produce TNF-α upon stimulation [30–32].

To study systemic inflammatory responses implicated in the pathogenesis of critical illness we measured a set of 15 biomarkers. Earlier investigations from our and other laboratories have used these biomarkers to obtain insight in host response disturbances in critically ill patients [2, 3, 33–35]. Especially biomarkers of systemic inflammation (CRP, IL-6, IL-8, MMP-8), endothelial activation (sICAM-1) and endothelial barrier dysfunction (angiopoietin 2/1 ratio) showed clear relationships with the extent of impairment of LPS-induced TNF-α production by blood leukocytes. This association was also present for coagulation activation, albeit to a lesser extent, as indicated by higher D-dimer and lower antithrombin levels in patients with the lowest TNF-α production capacity, while other coagulation parameters (platelet counts, PT, aPTT and protein C) were not different between groups. Of note, patients with a reduced TNF-α production capacity by whole blood leukocytes had a proportionally diminished capacity of blood leukocytes to produce IL-6, whilst IL-6 concentrations measured in (directly stored) plasma were proportionally increased. These seemingly opposing results can be explained by the fact that the whole blood stimulation assay measures IL-6 production of blood leukocytes stimulated by LPS, whilst plasma IL-6 levels reflect the resultant of IL-6 released into the circulation from a variety of (partially extravascular) cellular sources and the clearance of this cytokine from the circulation. We recently reported a study in patients with community-acquired pneumonia showing a similar association between a reduced capacity of blood leukocytes to produce proinflammatory cytokines upon ex vivo stimulation with LPS and stronger systemic proinflammatory responses relating to cytokine release, endothelial cell activation and activation of the coagulation system [36]. This investigation involved patients admitted to a general hospital ward and only a small subset had sepsis [36], suggesting that the association between immune suppression and hyperinflammation can also be detected in non-critically ill patients.

Our study has strengths and limitations. This investigation to the best of our knowledge for the first time addresses the association between immune suppression and systemic hyperinflammation in critically ill patients. We used unseparated blood leukocytes in a functional assay to measure immune suppression. The use of flow cytometry would have allowed for phenotypic characterization of specific leukocyte subsets, such as T and B cells. While the sample

size of our study is relatively small, our analyses did show strongly significant differences in systemic inflammatory responses between normal and low TNF-α producers. Our observational study does not address causal relationships between distinct host response aberrations. It should be emphasized that our study was not intended to generate information that could change clinical practice and/or could guide clinical decisions by physicians in the ICU. Rather, the results presented provide preliminary evidence that a commonly used feature of immune suppression in the ICU is associated with systemic responses that suggest concurrent hyperinflammation.

In critically ill patients the extent of immune suppression, as reflected by an impairment in the ability of blood leukocytes to produce proinflammatory cytokines upon stimulation, is proportional to the concurrent presence of systemic hyperinflammation. These data indicate that if one selects patients for immune stimulatory therapy based on a common readout such as the TNF-α production capacity of blood leukocytes, one likely also selects patients who have the strongest systemic inflammatory and endothelial cell responses. This knowledge is relevant for the development of precision medicine in critical care and selection of patients for treatment with immune stimulatory agents.

## Supporting information

**S1 Table. Admission diagnoses.** Abbreviations: COPD, chronic obstructive pulmonary disease.
(DOCX)

**S1 Fig. Whole-blood leukocyte responsiveness to LPS in critically ill patients and healthy volunteers.** Whole blood was drawn from 77 critically ill patients at 9:00 AM on the first day after admission to the ICU and from 19 age- and sex-matched healthy controls. Blood was stimulated for 3 hours with ultrapure LPS (100 ng/mL), and tumor necrosis factor (TNF)-α and interleukin (IL)-1β, and IL-6 concentrations were measured in supernatants. Data are presented box and whisker diagrams as specified by Tukey. HV, healthy volunteers; ICU, critically ill patients. ***P < 0.001, ****P < 0.0001.
(TIF)

**S2 Fig. Whole-blood leukocyte tumor necrosis factor-α production in response to LPS in critically ill patients adjusted for monocyte count.** Whole blood from critically ill was stimulated for 3 hours with ultrapure LPS (100 ng/mL). Tumor necrosis factor (TNF)-α concentration was measured in supernatants. TNF-α concentrations per $10^6$ monocytes in whole blood are stratified according whole blood TNF-α production capacity (i.e. quartiles of TNF concentration in supernatants after LPS stimulation). Data are presented as box and whisker diagrams as specified by Tukey, in 59 patients in whom white blood cell differentials were available. **P < 0.01, ***P < 0.001, ****P < 0.0001.
(TIF)

**S1 Data.**
(CSV)

**S2 Data.**
(CSV)

## Acknowledgments

The authors would like to thank the members of the BASIC study group: Friso M. de Beer, Lieuwe D. J. Bos, Gerie J. Glas, Roosmarijn T. M. van Hooijdonk, Janneke Horn, Tom van der

Poll, Laura R. A. Schouten, Marcus J. Schultz, Marleen Straat, Lonneke A. van Vught, Luuk Wieske, Maryse A. Wiewel, and Esther Witteveen, as well as all subjects who participated in this study, and Barbara S. Dierdorp and Tamara Dekker for their help with the workup of the cytokine and plasma biomarker measurements.

## Author Contributions

**Conceptualization:** Xanthe Brands, Fabrice Uhel.

**Data curation:** Xanthe Brands, Fabrice Uhel, Maryse A. Wiewel, René Lutter.

**Formal analysis:** Tom van der Poll.

**Funding acquisition:** Xanthe Brands, Tom van der Poll.

**Investigation:** Xanthe Brands, Lonneke A. van Vught, Arie J. Hoogendijk.

**Methodology:** Xanthe Brands.

**Project administration:** Xanthe Brands.

**Supervision:** Tom van der Poll.

**Writing – original draft:** Xanthe Brands, Fabrice Uhel.

**Writing – review & editing:** Xanthe Brands, Fabrice Uhel, Lonneke A. van Vught, Maryse A. Wiewel, Arie J. Hoogendijk, René Lutter, Marcus J. Schultz, Brendon P. Scicluna.

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
