## [Decision Letter · Decision Letter 0]

28 Mar 2022

PONE-D-21-33755Immune suppression is associated with enhanced systemic inflammatory, endothelial and procoagulant responses in critically ill patientsPLOS ONE

Dear Dr. Xanthe Brands ,

Thank you for submitting your manuscript to PLOS ONE. After careful consideration, we feel that it has merit but does not fully meet PLOS ONE’s publication criteria as it currently stands. Therefore, we invite you to submit a revised version of the manuscript that addresses the points raised during the review process. I would appreciate if you pay a careful attention in your response to the reviewer's comments. 

We look forward to receiving your revised manuscript.

Kind regards,

Ehab Farag, MD FRCA FASA

Academic Editor

PLOS ONE

Journal Requirements:

2. We noted in your submission details that a portion of your manuscript may have been presented or published elsewhere. Please clarify whether this publication was peer-reviewed and formally published. If this work was previously peer-reviewed and published, in the cover letter please provide the reason that this work does not constitute dual publication and should be included in the current manuscript.

Reviewers' comments:

Reviewer's Responses to Questions

**Comments to the Author**

1. Is the manuscript technically sound, and do the data support the conclusions?

Reviewer #1: Partly

2. Has the statistical analysis been performed appropriately and rigorously? 

Reviewer #1: Yes

3. Have the authors made all data underlying the findings in their manuscript fully available?

Reviewer #1: No

4. Is the manuscript presented in an intelligible fashion and written in standard English?

Reviewer #1: Yes

5. Review Comments to the Author

Reviewer #1: This manuscript investigates immune biomarkers in plasma correlated with inflammatory pathways using patients admitted to ICU. The statistical analysis methods are fine for the presented data analysis. I have below questions and comments.

The objective of this study was to determine the relationship between the degree of immune suppression and systemic inflammation in patients shortly after admission to the ICU. Was the TNF-a used to identify the degree of immune suppression?

It is an observational study, but it is not clear how the sample size was decided. Please provide sample size considerations.

Table 1 reported the distribution of the stratification TNF-a across infectious diseases. What’s the distribution of the stratification TNF-a across noninfectious diseases? How many noninfectious diseases also have signs of immune suppression?

This study examined the correlations between TNF-a and other immune biomarkers by comparing immune biomarkers among quartile stratified group of TNF-a. Spearman’s r was also used for a couple of markers. For these correlations, would patient’s characteristics (e.g. age, sex, BMI, etc.) mediate the relationship between TNF-a and other immune biomarkers?

6. PLOS authors have the option to publish the peer review history of their article (what does this mean?). If published, this will include your full peer review and any attached files.

Reviewer #1: No

---

## [Author Response · Author response to Decision Letter 0]

14 Jun 2022

Reviewer This manuscript investigates immune biomarkers in plasma correlated with inflammatory pathways using patients admitted to ICU. The statistical analysis methods are fine for the presented data analysis. I have below questions and comments.

Comment #1: The objective of this study was to determine the relationship between the degree of immune suppression and systemic inflammation in patients shortly after admission to the ICU. Was the TNF-a used to identify the degree of immune suppression?

Response:

We thank the reviewer for his/her insightful comments. 

Indeed, as specified in the results section (page 7, lines 143-146, 152-153): “Given that a reduced TNF-α production capacity by blood leukocytes has been widely recognized as a hallmark of immune suppression in critically ill patients, we stratified patients into four groups based on quartiles of LPS-induced TNF-α production. […] the stratification of patients based on TNF-α production capacity of blood leucocytes resulted in conditions of increasing degrees of immune suppression.”

In other words, the identification of immune suppression was based on a reduced ability of blood leukocytes to produce TNF-α in response to LPS-stimulation. 

In order to further clarify this readout, we rephrased the objective of the study in the introduction: 

Introduction (Pages 3, 4): “To this end, we used the decreased capacity of whole blood leukocytes to produce TNF-α in response to LPS-stimulation as a readout for critically-ill patient immune suppression”

Comment #2: It is an observational study, but it is not clear how the sample size was decided. Please provide sample size considerations.

Response:

We did not perform a formal sample size calculation prior to this study, since to the best of our knowledge previous studies testing the association between whole blood leukocyte stimulations to biomarkers of systemic inflammation have not been performed. Thus, our study provides preliminary estimates of variance, deviations etc.. for the design of future specific studies. This is now commented upon in the statistical paragraph as follows:

Methods (page 6): “A formal sample size calculation was not done prior to the study (to the best of our knowledge previous studies associating whole blood leukocyte stimulations with biomarkers of systemic inflammation have not been performed)” 

Comment #3: Table 1 reported the distribution of the stratification TNF-a across infectious diseases. What’s the distribution of the stratification TNF-a across noninfectious diseases? How many noninfectious diseases also have signs of immune suppression?

Response:

Table one shows the distriubution of TNF-α production capacity in all patients admitted to the ICU, including patients admitted for an infectious(n=51) and for a non-infectious diagnosis (n=26). Of these, 40 (78.4%) patients with a sepsis admission diagnosis and 18 (69.2%) patients with a non-infectious admission diagnosis showed reduced TNF production capacity (≤896 pg/mL), consistent with immune suppression. 

In order to clarify this point, we added the following to the manuscript: 

Results (page 8): “Among patients admitted for a sepsis (n= 51) or for a non-infectious diagnosis (n= 26), 40 (78.4%) and 18 (69.2%) showed reduced TNF production capacity (≤896 pg/mL), respectively.”

Comment #4: This study examined the correlations between TNF-a and other immune biomarkers by comparing immune biomarkers among quartile stratified group of TNF-a. Spearman’s r was also used for a couple of markers. For these correlations, would patient’s characteristics (e.g. age, sex, BMI, etc.) mediate the relationship between TNF-a and other immune biomarkers? 

Response:

As shown in table 1, no association was found between TNF-α production capacity and patient age, sex or BMI. We therefore considered it unlikely that these clinical parameters mediate the association between TNF-α production capacity and other plasma biomarker concentrations.

---

## [Editor Report · Decision Letter 1]

6 Jul 2022

Immune suppression is associated with enhanced systemic inflammatory, endothelial and procoagulant responses in critically ill patients

PONE-D-21-33755R1

Dear Dr. Xanthe Brands ,

We’re pleased to inform you that your manuscript has been judged scientifically suitable for publication and will be formally accepted for publication once it meets all outstanding technical requirements.

Kind regards,

Ehab Farag, MD FRCA FASA

Academic Editor

PLOS ONE
---

## [Editor Report · Acceptance letter]

15 Jul 2022

PONE-D-21-33755R1 

Immune suppression is associated with enhanced systemic inflammatory, endothelial and procoagulant responses in critically ill patients 

Dear Dr. Brands:

I'm pleased to inform you that your manuscript has been deemed suitable for publication in PLOS ONE. Congratulations! Your manuscript is now with our production department. 

Kind regards, 

on behalf of

Dr. Ehab Farag 

Academic Editor

PLOS ONE